# Prevalence of Asthma and Its Associating Environmental Factors among 6–12-Year-Old Schoolchildren in a Metropolitan Environment—A Cross-Sectional, Questionnaire-Based Study

**DOI:** 10.3390/ijerph182413403

**Published:** 2021-12-20

**Authors:** Dávid Molnár, Gabriella Gálffy, Alpár Horváth, Gábor Tomisa, Gábor Katona, Andor Hirschberg, Györgyi Mezei, Monika Sultész

**Affiliations:** 1Department of Anatomy, Histology and Embryology, Semmelweis University, 58 Tűzoltó Street, H-1094 Budapest, Hungary; 2Medical Centre, Department of Otorhinolaryngology and Head and Neck Surgery, Hungarian Defence Forces, 109-111 Podmaniczky Street, H-1062 Budapest, Hungary; 3Pest County Pulmonology Hospital, Törökbálint, 70 Munkácsy Mihály Street, H-2045 Torokbalint, Hungary; ggalffy@hotmail.com (G.G.); a.horvath@chiesi.com (A.H.); 4Medical Department of Chiesi Hungary Ltd., 2 Dunavirág Street, H-1138 Budapest, Hungary; g.tomisa@chiesi.com; 5Department of Oto-Rhino-Laryngology, Heim Pál National Pediatric Institute, 86 Üllői Street, H-1089 Budapest, Hungary; katonagbor@gmail.com (G.K.); sultmon@gmail.com (M.S.); 6Department of Oto-Rhino-Laryngology and Maxillo-Facial Surgery, Saint John’s Hospital, 1-3 Diós árok Street, H-1125 Budapest, Hungary; andor.hirschberg@janoskorhaz.hu; 71st Department of Paediatrics, Division of Allergo-Pulmonology, Semmelweis University, 53–54 Bókay János Street, H-1083 Budapest, Hungary; mezei.gyorgyi@med.semmelweis-univ.hu

**Keywords:** asthma, schoolchildren, prevalence, nutrition, environment, lifestyle, atopy, allergy, east-central Europe, ISAAC

## Abstract

We aimed to evaluate the prevalence of asthma and its associating environmental factors within a 6–12-year-old population. A cross-sectional, questionnaire-based study was conducted in primary schools located in the capital of Hungary; 3836 eligible parent-reported questionnaires were evaluated. Besides the International Study of Asthma and Allergies in Childhood (ISAAC) phase three core questions for asthma, the survey also assessed various potential risk factors. We introduced the umbrella term cumulative asthma as the union of physician-diagnosed asthma and current wheezing to estimate the lifetime prevalence of asthma. Current wheezing and physician-diagnosed asthma showed a frequency of 9.5% and 6.3%, respectively. They contributed to a cumulative asthma prevalence of 12.6% among the sampled population, with a girl-boy percentage of 37.4% to 62.6%. Air-pollution and weedy areas were associated with greater risk for asthma, while a suburban residence showed lesser odds. Indoor smoking, visible mold, and keeping a dog were defined as risk factors for asthma, while the presence of plants in the bedroom and pet rodents were associated with lower odds ratios. The consumption of fast food, beverages containing additives and margarine were significantly higher in asthmatics, while we found frequent sport activity and cereal intake associated with lower odds ratios for asthma. In this urban environment, we identified an increased asthma prevalence compared to some previously published studies, but the cross-sectional design and the different methodology did not permit us to draw timeframe-dependent conclusions.

## 1. Introduction

Asthma is a heterogeneous disease with time-varying respiratory symptoms (including wheeze) and is often associated with airway hyperresponsiveness and inflammation [1]. Taking the history of the characteristic symptoms is the key to the diagnostic work-up, reinforced with evidence of fluctuating airflow limitation. Questionnaire-based studies are amenable tools to estimate asthma prevalence, especially for large-scale surveys in pediatric focus groups, with whom a physician’s office visit is impractical [2]. The International Study of Asthma and Allergies in Childhood (ISAAC) included three research phases between 1991 and 2012 and developed various questions to investigate asthma and allergy [3]. The rationale behind the ISAAC initiative was to conduct epidemiologic research concerning the more recent symptoms that covered the past 12 months, to minimize recall errors [1]. The core questions of the ISAAC surveys are still favorable tools for such epidemiologic research in respiratory medicine.

Our questionnaire-based study was conducted in Budapest (Hungary). We embedded the ISAAC core questions into a form to evaluate the current asthma prevalence in the 6–12-year-old population. Inner cities are the areas where asthma is more frequent. Urban air pollution has been implicated as one of the factors responsible for the dramatic increase in asthma incidence. Our goal was to assess the prevalence of asthma in our metropolitan area and present the data in a representative survey with a large number of participants almost two decades after the initiation of phase three of the ISAAC project [3].

A secondary aim was to reveal different environmental factors and habits associated with asthma in the region of interest. We examined the association of various indoor and outdoor pollutants (e.g., exposure to industrial air pollution and smoking, exposure to pets), the consumption frequency of nutrients and physical activity,

## 2. Materials and Methods

### 2.1. Ethical Considerations

The study was endorsed by the Ethics Committee of the Heim Pál National Pediatric Institute, Budapest (KUT-19/2019). Designing the study and collecting, handling and processing the scientific data was carried out according to the principles of the Helsinki Declaration. Informed consent was obtained from all responders.

### 2.2. Study Design

This cross-sectional, questionnaire-based study was carried out in September 2019. A total number of twenty-one primary schools in eight districts of Budapest were randomly selected from the listings provided by the Central Data Processing and Registration Office of the Hungarian Ministry of Interior. Parents of 6–12-year-old children, at the first teacher-parent meetings of the school year, were asked to complete the survey. The teachers provided detailed instructions before completion. The questionnaires were collected at the end of teacher-parent meetings or within a week timeframe.

### 2.3. The Questionnaire

The enrolled parents received the multi-aspect questionnaire. In the present study, we summarized the results related to asthma and its putative risk factors. To address the prevalence of asthma and its risk factors, we included the core questions for asthma according to the phase three manual of the ISAAC supplemented with self-designed queries on physician-diagnosed asthma and risk factors [4].

Subjects who responded “Yes” to the ISAAC core question “Has your child had wheezing or whistling in the chest in the past 12 months?” constituted the “current wheezing” group (CW). The prevalence of “physician-diagnosed asthma” (PDA) was determined based on the answers to the question “Has your child had asthma diagnosed by a physician?”. The union of CW and PDA sets defined the “cumulative asthma” group (CA) (Figure 1), best approximating the concept of lifetime asthma among the focused age group of pupils.

We also included questions dealing with associated atopic conditions (A), including food allergy, asthma and allergic rhinitis.

Environmental (E), lifestyle (L) and nutritional (N) factors were evaluated in association with the CA group. The presence of a given environmental factor was addressed in the form of a yes-no question (E01-E34, Appendix A). ZIP codes were registered to distinguish whether a student had a residence in the capital or the suburbs. The frequency of lifestyle factors was determined by questions expecting answers to three predefined choices (L01–L02), except for evaluating smoking behavior (L03), when a polar question was used. The frequency of weekly consumption of different nutrients was initially planned to be evaluated upon three predefined choices: “≥3 times”, “1–2 times”, “Rarely [<1 times]” (N01–16). Despite the given options, parents often responded with other intervals (e.g., 2–3 times a week) covering two choices. To overcome this, answers were rearranged into two intervals during evaluation: “Frequently [≥1 times per week]” or Rarely [<1 times per week]. These derivative answers were marked with the letter “d” in the report (N01d-N16d, Appendix A). We asked for anthropometric data of the subjects, including heights and weights, and age and gender related body mass index (BMI) percentiles (BMI-for-age percentiles) were calculated according to the guideline and reference tables of the Association of Hungarian Health Visitors [5].

### 2.4. Statistical Analysis and Data Visualization

The data was characterized by standard descriptive statistics: frequencies (percentages) and means for categorical and quantitative data, respectively. Binomial logistic regression and chi-square were used to compare frequencies, and the t-test was used to compare the means of groups. Individual regression models were applied for the different explanatory variables. The response variable was cumulative asthma in all models. There was no reference group in the case of BMI percentages, so we used the chi-squared test only. Results were considered statistically significant at *p* < 0.05. In the case of categorical variables, odds ratios (OR) and 95% confidence intervals (95% CI) were calculated. Prevalence estimates were calculated by dividing positive responses to the given question by the total number of completed questionnaires. Percentages were calculated by dividing the frequency by the total number of observations, excluding missing answers, and then multiplying by 100. All analyzes were performed with the R-3.6.2 for Windows statistical program software [6].

## 3. Results

### 3.1. Data Acquisition, Refinement and Demographic Parameters of the Study Group

A total of 6869 questionnaires were distributed in the 21 primary schools. Altogether 3885 forms were returned, of which 49 had to be ruled out due to technical reasons or the subject’s inappropriate age. Of the 3836 eligible students 51.6% (*n* = 1979) were girls and 48.4% (*n* = 1857) were boys. The mean age of the children was 10.33 years +/− 1.68 (Figure 2).

### 3.2. Prevalence of Asthma

In the last 12 months according to the ISAAC core questions, 9.3% of responders (*n* = 356) experienced wheezing. This set of subjects represented the current wheezing (CW) group. The symptoms’ frequency and characteristics are also included in Table 1. Physician-diagnosed asthma (PDA) was 6.5% (*n* = 248) of the pupils regardless of being symptomatic or not in the previous year (Table 1 and Table 2). Thus, cumulative asthma (CA) had a prevalence of 12.6% (*n* = 484) among the 6–12-year-old schoolchildren in the sampled environment (Table 2). Among girls, asthmatics had a significantly lower proportion (OR = 0.52, CI: 0.42–0.63, *p* < 0.0001), and girl-boy percentage was 37.4% (*n* = 181) to 62.6% (*n* = 303) in the CA group.

### 3.3. The Severity of Asthma and Associating Atopic Diseases

The ISAAC core questions also describe the severity of a subject’s disease. Table 3 demonstrates the frequency of such symptoms (including sleep disturbances, speech limiting wheezing and wheezing related to physical exercises) in the CA group.

Eczema, food allergy and allergic rhinitis had significant associations with CA (Table 4). Physician-diagnosed allergic rhinitis had the greatest odds for asthma (OR = 6.21, CI: 4.90–7.86, *p* < 0.0001) out of this sequence.

### 3.4. Environmental Risk Factors

Thirty-four questions aimed to analyze any possible relationships between the prevalence of cumulative asthma and local environmental risk factors (Appendix A). Figure 3 visualizes the statistical association between CA and the risk factors.

The odds of the cumulative asthma group were lower among schoolchildren who lived in the suburbs than those who had their residence in the capital (OR = 0.64, CI: 0.43–0.93, *p* = 0.0236).

Among indoor causative agents, smoking at home had a crucial role in the prevalence of asthma. Whether the child had been exposed during the first year of life (OR = 1.61, CI: 1.18–2.16, *p* = 0.0018) or at the time of the survey (OR = 1.65, CI: 1.24–2.18, *p* = 0.0005), smoking associated with asthma. Visible mold (OR = 2.13, CI: 1.36–3.24, *p* = 0.0006) in the children’s bedroom contributed to the occurrence of asthma, though, the presence of a plant (OR = 0.79, CI: 0.65–0.96, *p* = 0.0191) yielded lower odds ratio. Dog (OR = 1.44, CI: 1.13–1.82, *p* = 0.0029) keeping associated with CA, while rodent ownership (OR = 0.72, CI: 0.52–0.98, *p* = 0.0454) showed lower odds ratio. On the contrary, cat, bird, or furry animal keeping did not.

The proximity of any air-polluting factories (OR = 1.33, CI: 1.05–1.68, *p* = 0.0161), heavy-vehicle traffic (OR = 1.29, CI: 1.05–1.59, *p* = 0.0161) or a weedy area (OR = 1.39, CI: 1.15–1.69, *p* = 0.0007) had a significant association with the prevalence of CA.

### 3.5. Lifestyle Factors

Sports activity in a frequent manner (≥3 times a week) was significantly associated with the absence of asthma (OR = 0.69, CI: 0.52–0.92, *p* = 0.0098) (Table 5). Visual display time did not associate with the prevalence of CA. Although environmental tobacco smoke increased asthma risk (OR = 1.65, CI: 1.24–2.18, *p* = 0.0005), pupils who smoked showed no significant association with CA. The ratio of smokers among students was just 1.04% (*n* = 4) though.

### 3.6. Nutritional Factors

The weekly consumption of a particular ingredient was derived from the original questions as per the method section (Appendix A). Frequently eating fast food (e.g., hamburgers) or drinking soft drinks including artificial additives was associated with the prevalence of CA (OR = 1.75, CI: 1.39–2.19, *p* < 0.0001 and OR = 1.27, CI: 1.05–1.53, *p* = 0.0157 respectively). Weekly consumption of margarine had a significant association with the development of asthma (OR = 1.35, CI: 1.11–1.65, *p* = 0.0026). The intake of cereals at least once a week seemed preventive for asthma (OR = 0.54, CI: 0.41–0.73, *p* < 0.0001) (Figure 4).

Based on the available data, we were able to calculate the BMI-for-age percentile of a child in the case of only 3295 participants. Among them, we identified an association (Chi-square = 16.26 df = 5 *p* = 0.0062) between the percentile values and cumulative asthma (Figure 5, Appendix A). Children with higher BMI were more likely to have asthma.

## 4. Discussion

In the present study the prevalence of current wheezing based on the corresponding ISAAC core question “Has your child had wheezing or whistling in the chest in the past 12 months?” was 9.3% among the 6–12-year-old children in the metropolitan area of Budapest, the capital of Hungary. Physician-diagnosed asthma accounted for 6.5% of prevalence. Union of these two sets of students defined a cumulative asthma prevalence at 12.6%. Hungary participated with two centers in the phase three study of ISAAC in 2003, but none represented the capital. A survey (ISAAC ID 047002) covering schools of two cities from the southeastern lowland of Hungary involved solely the 13–14 age group. Out of the 2899 students who had completed the written questionnaire 204 (7.1%) reported wheezing in the previous year [7]. The other study (ISAAC ID 047001) indicated a 6.6% and 5% prevalence of current wheeze in the 6–7-year-old and 13–14-year-old groups respectively [7]. The overall higher prevalence of the contemporary sampling frame’s current wheezing can be either due to e.g., the urban environment or a gradually increasing prevalence in the fifteen-year timeframe.

We found cumulative asthma significantly lower among girls than in boys. There are controversial results regarding the sex-specific prevalence of childhood asthma, especially in a lifetime analysis. However, current symptom prevalence has been reported with an increasing boy-girl ratio [8].

From a Central European perspective, the prevalence of asthma shows heterogeneity. Aberle et al. reported 7.9% of current wheezing and 4.1% of diagnosed asthma among 10–11-year-old students in neighboring Croatia [9]. Their survey provided evidence that more boys (63.2%) had asthma. The gender-specific relations are consistent with our results. However, the geographical features of the investigated area resembled Hungary, the level of urbanization did not approximate Budapest, which can explain the higher prevalence in our settings. Results of the Croatian study are more comparable with and more congruent to the former ISAAC surveys conducted in Hungary (ISAAC ID 047001 & 047002) [7].

An almost population-wide screening of primary school students was carried out in Tyrol, Austria [10]. The mean age of the participants was 8.4 years (SD ± 1.2). Besides the current wheezing, defined by the ISAAC manual, they used an extended definition for asthma by massing subjects with current wheeze or who had used an asthma spray ever or recurrent wheezy bronchitis or a doctor diagnosis. 10.3% of the total study population had current wheezing, which is 1% higher than in our subjects. Doctor-diagnosed asthma was at 3.4%. Based on their residence, Tyrolean students were sub-classified into farm children, rural children and Innsbruck-town children, and their exposure to particular agents (e.g., hay-loft, animal shed, or farm milk). They found living on a farm as protective but only for those with regular exposure to farming agents. In our study, children commuting from the suburbs had a lower risk of asthma. With no history of exposure to particular agents, 21.3% of Innsbruck-town children fulfilled the extended criteria of asthma. This is 1.70 times higher than our cumulative asthma results, which can be due to either the regional differences or the different sampling protocols.

In a study from Romania, asthma-like symptoms (dominantly dry cough) were reported among 20% of the participating students [11]; 48% of the subjects were exposed to environmental tobacco smoke, one of the expected triggers responsible for the high rate of symptoms.

Out of our participants, 9.65% (*n* = 370) shared their home with a smoker relative, which is a remarkable difference compared to the Romanian data. Sixty-eight (18.15%) out of them fit into the CA group and verified tobacco exposure as a major risk factor for asthma. Indoor tobacco fumes also impact other respiratory outcomes in children, including pneumonia, night cough and croup [12].

Increasing evidence has confirmed mold as another predisposing factor in children [13,14,15,16,17]. Visible mold can be a source of fungal spores and other volatile organic compounds. These viable and non-viable particles are sufficient enough to increase the risk of asthma [15]. In our sample, moldy surfaces doubled the chance of asthma development. Fagbule and Ekanem found that mold was only harmful in the bedroom, and on the contrary, it had a somehow protective effect elsewhere at home [18]. The authors noticed that children could also benefit from indoor plants [18]. Our study also revealed this favorable correlation between the presence of plants in the bedroom and asthma prevalence. It is challenging to explain this phenomenon because potted plants’ soil could serve as an origin for fungal and other biological agents.

The literature is controversial on the role of pet allergens in asthma prevalence [19,20,21]. We found that overall pet ownership did not associate with asthma in accordance with current meta-analyses [21]; however, pet-specific risks differed. A UK birth cohort showed that early childhood ownership could have a prophylactic effect but surrounding rabbits or rodents could contribute to non-atopic asthma with a higher odds [22]. Exposure to mouse antigens could associate with wheezing [23]. Among the subjects of the current study, a rodent’s presence resulted in lower odds of asthma. Inconsistency might be a result of a non-species-specific investigation. In addition, rodents were subjects of leisure pet ownership in our study, but the exposure to their antigens can also be from pests invading households. Such circumstances may also associate with other pollutive agents and a lower socioeconomic environment. Our observations concluded dog-keeping as a potential risk factor of asthma, but cat-ownership did not. An explanation could be the so-called cat paradox [22,24,25]. The increased odds associated with dog ownership argue with the contemporary view [21]. The latter statistical relationship seems plausible but might be casual; a more detailed assessment of pet ownership should be conducted in the future.

Outdoor air pollution also plays a role in asthma development. In the current setting, the proximity of any air-polluting factories or heavy vehicle traffic is associated with the risk of asthma. Although numerous reports support these findings, from an environmental health point-of-view the particular composition and concentration of airborne pollutants instead reflect the association [26,27,28,29].

The protective effect of suburban living can also be a surrogate reference of this association between air pollution and asthma because exposure to ubiquitous atmospheric agents is suspected to be lower at those sites.

Living in a highly weeded area is considered a risk factor of asthma development with an OR of 1.33. It is also associated with allergic rhinitis prevalence in the same environment, where common ragweed *Ambrosia artemisiifolia* is the most widespread cause of allergy-associated symptoms [30]. However, a higher level of pollen load showed a non-significant association with allergy risk in Budapest [31]. Despite the geographical distance, it is worth noting that 14% of asthmatic children of the 3–11-year-old population sensitized against ragweed in a US-based study [32].

Measurement of physical activity is challenging to carry out: besides the duration and frequency, the intensity is supposed to be a question of interest. Questionnaire-based research is a limiting factor of good quality and comparable data. Our current survey reported a significant association when children realizing vigorous activity more than three times a week were less likely to be in the cumulative asthma group. Visual display time as a reference for sedentary lifestyle correlated with the odds ratio of asthma, but the relationship was not statistically significant.

The ISAAC Phase Three summarized similar tendencies, but the methodology was different [33]. Other studies with different approaches reported an ambiguous correlation between physical activity and asthma prevalence [34,35,36,37,38]. We must emphasize that asthma has also been reported as a barrier to physical activity; thus, in the future, we should analyze this relation through an interdisciplinary lens [34]. Emerging technologies, including wearable biosensors or smartphone applications, may serve as more accurate physical activity data resources in the future.

Dietary patterns influence the risk of asthma development. A Mediterranean-style diet, rich in fruits, vegetables and whole grains while taking less meat and dairy in, and other plant-based foods have associated with reduced odds for asthma [39,40,41,42,43]. A Westernized diet, including a predominant amount of animal products with higher fat intake and lower fiber consumption, is a risk factor for asthma [41]. It would be welcome to keep adherence to a healthier diet where it is already historically and geographically predisposed; trends showed an increased prevalence of asthma symptoms in the Mediterranean and Latin America [41,44]. In the current sampled population, such emblematic meals of urbanized living like fast-food and drinks with artificial additives are associated with cumulative asthma.

Regular consumption of margarine also showed an association with cumulative asthma. A high rate of fat intake is also attributed to the Westernized diet, but there is more emphasis placed on the composition of fat fractions. According to the lipid hypothesis, an increased intake of polyunsaturated fatty acids (PUFAs) over saturated fat can increase asthma prevalence and allergic sensitization, but controversial results are also available [41,42,45,46]. Margarine can serve as a significant source of PUFAs. Therefore, we should be aware of its potential causative role in asthma and atopy development [45,47,48,49].

Frequent intake of cereals showed an association with decreased odds of cumulative asthma. The literature has supported this finding, though the protective mechanism is not completely understood yet [41,42,43,46]. Our survey did not evaluate prior cereal consumption, although the early introduction of cereal grains into the diet is substantial to avoid sensitization [50].

High energy diet is a predisposing factor for obesity. Obesity is associated with worse asthma control and an increased risk of exacerbations in all ages [41]. Besides the fact that obese and overweight children are more likely to have asthma, according to the current view, obesity-related asthma in childhood is a separate entity with a Th1 cell polarization [41,51,52]. Our observation reinforced the conception of the association between asthma and BMI-for-age percentiles, but the phenotyping of obese asthmatics was beyond our scope.

## 5. Limitations and Strengths

Some limitations should be considered when interpreting our results. First, the prevalence of asthma was measured based on questionnaires, without performing clinical tests. Second, such a survey cannot collect data regarding the demographic and health features of non-responders. Members of the focus group can also differ in health literacy which can either influence the understanding of the questions and reporting current prior and current medical conditions.

Current wheezing symptoms were related to the past 12 months, which may involve further study limitations. On the other hand, the reported current wheeze supports defining a group of children who have not been diagnosed with asthma symptoms due to undisclosed reasons. The cumulative asthma group was set up for a better approximation of the composition of childhood asthma prevalence. Thus, the outcomes may have been either over or underestimated.

The food consumption frequency results refer to the one-year pre-survey period. Keeping in mind the wide age-group of the subjects, this is a short period to define putative etiologic relationships. Longitudinal surveys or twin studies would be more appropriate tools.

The strengths of this study are that we used a standardized method and expanded the questionnaire with numerous environmental and lifestyle-related questions, stressing the main environmental factors associated with asthma. The database is large enough to reflect the population of Budapest. We added comparable considerations of the associating factors. Our cross-sectional survey can serve as a basis for further longitudinal studies to have a clearer view of the prevalence and characteristics of childhood asthma.

## 6. Conclusions

Our study delivered data on the recent prevalence of asthma among 6–12-year-old children in Budapest, Hungary. We found the prevalence of current wheezing 9.3% among the subjects, in comparison 6.5% had already been diagnosed with asthma by a physician. We introduced the term “cumulative asthma” to estimate the lifetime prevalence of asthma. This group included 12.6% of the individuals.

In this urban environment, we identified an increased asthma prevalence compared to some previously published studies, but the cross-sectional design and the different methodology did not permit us to draw timeframe-dependent conclusions.

Besides the outdoor and indoor environmental factors, we also analyzed the contribution of lifestyle and nutritional determinants. Suburban residence, plant in the bedroom, rodent ownership, frequent consumption of cereals and regular sport activity are associated with lower odds ratios for asthma. In comparison, nearby heavy vehicle traffic or air polluting factories, weedy areas, visible mold, indoor smoking, dog ownership, the consumption of fast food, beverages containing additives and margarine, and high BMI showed significant association with asthma.

Though the statistical associations revealed are not numerous, the majority fit in the existing literature. The controversial impact of rodent and dog ownership requires a better understanding.

Our study serves large-scale descriptive data on the local prevalence of asthma which has not been examined in the capital area. Novel devices and prospective, interventional studies with interdisciplinary approaches may help us reduce the burden of asthma by comprehending its triggers.

## Figures and Tables

**Figure 1 ijerph-18-13403-f001:**
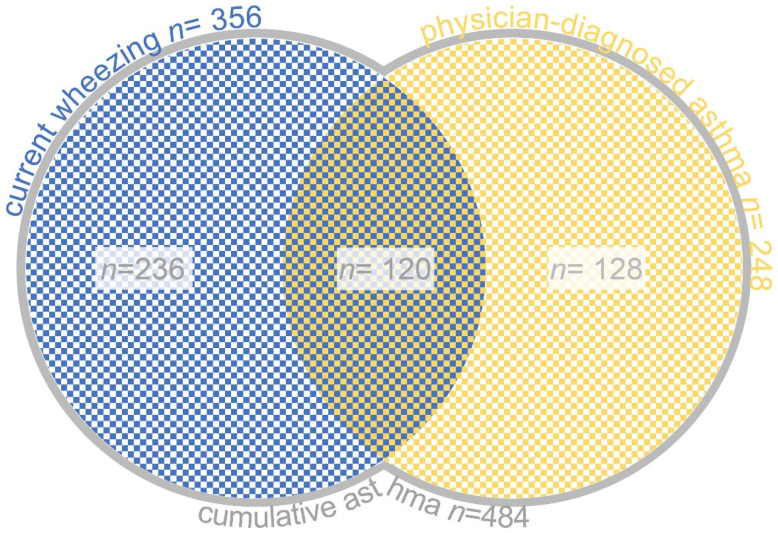
Definition of cumulative asthma: the union of current wheezing and physician-diagnosed asthma sets.

**Figure 2 ijerph-18-13403-f002:**
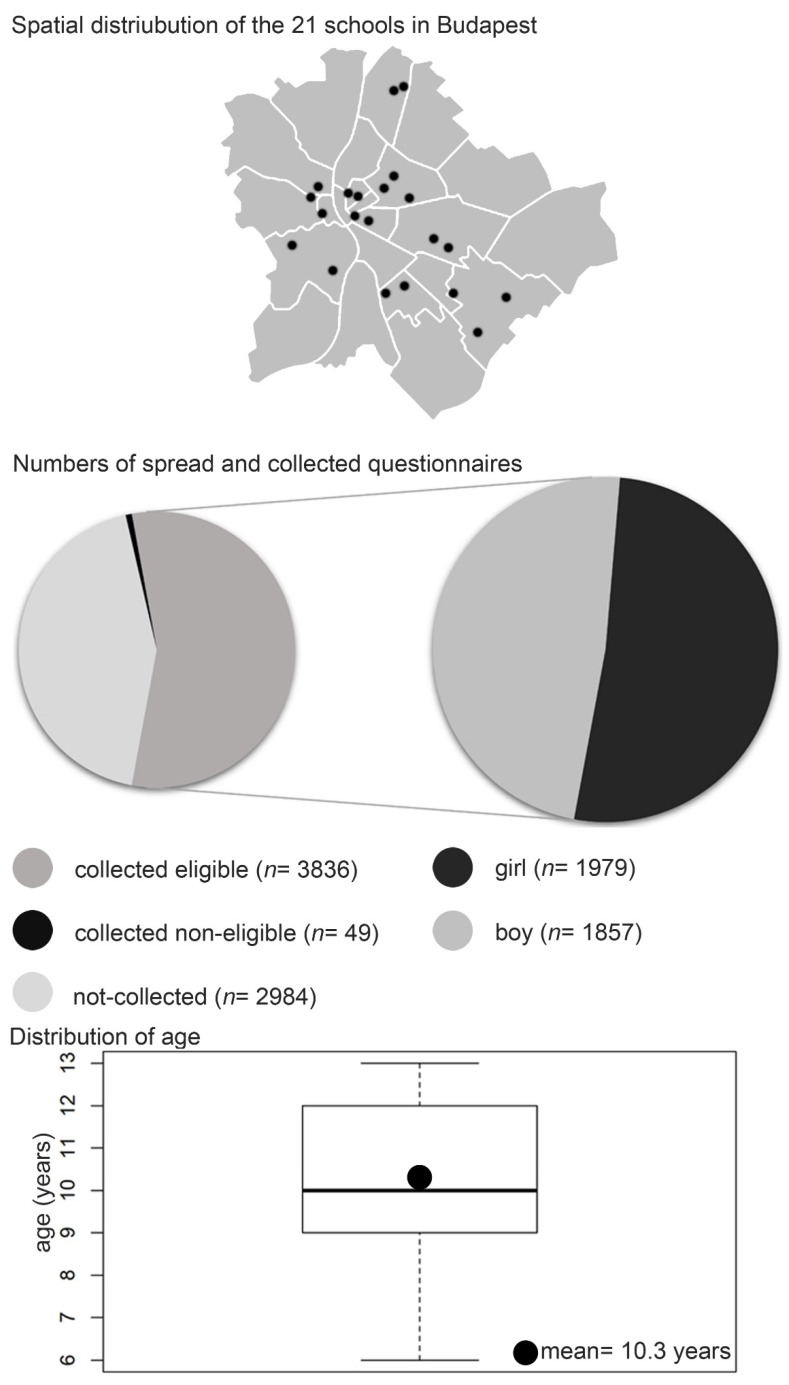
The schematic map of Budapest demonstrates the distribution of the selected primary schools. The Pie chart demonstrates the ratio of the collected forms and the gender-specific rate of the eligible.

**Figure 3 ijerph-18-13403-f003:**
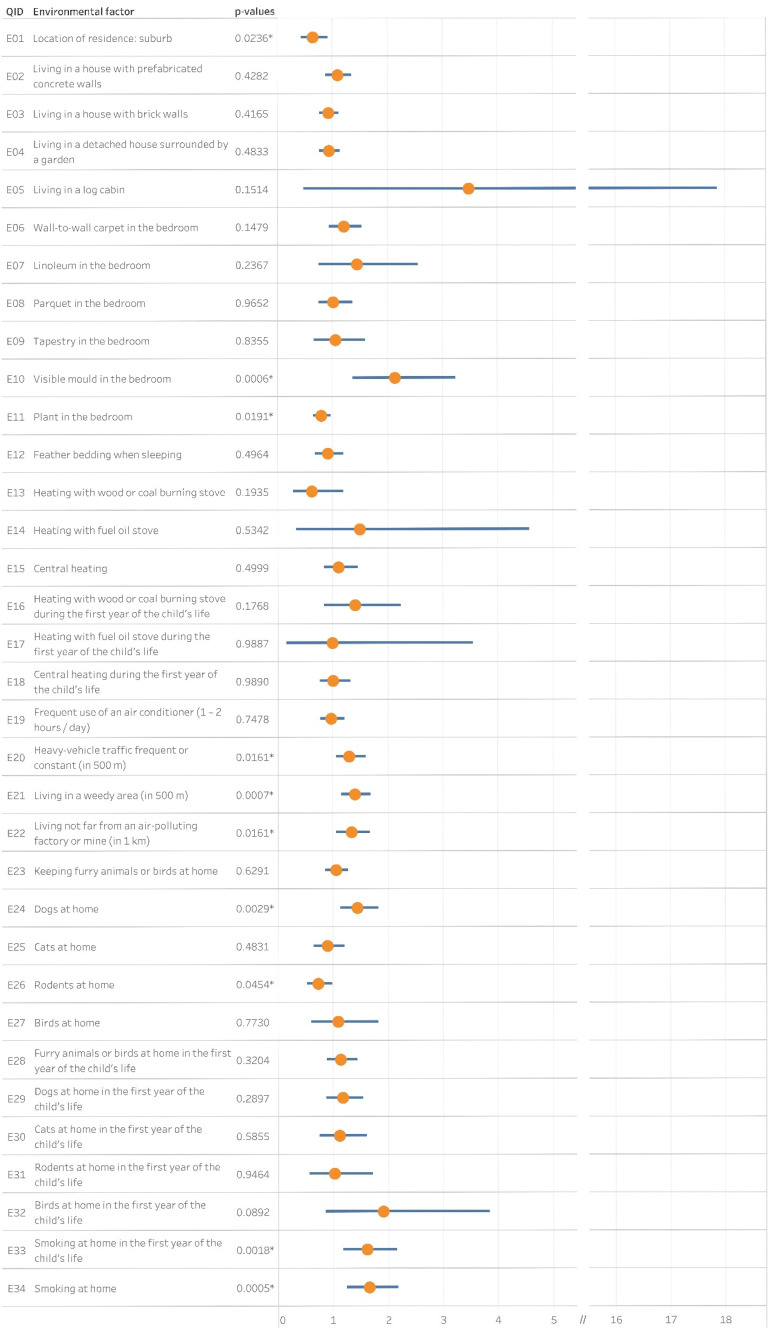
The association between cumulative asthma and indoor and outdoor environmental factors. Yellow dots and blue bars stand for the corresponding odds ratios and confidence intervals, respectively. QID: Question ID, *: *p* <0.05.

**Figure 4 ijerph-18-13403-f004:**
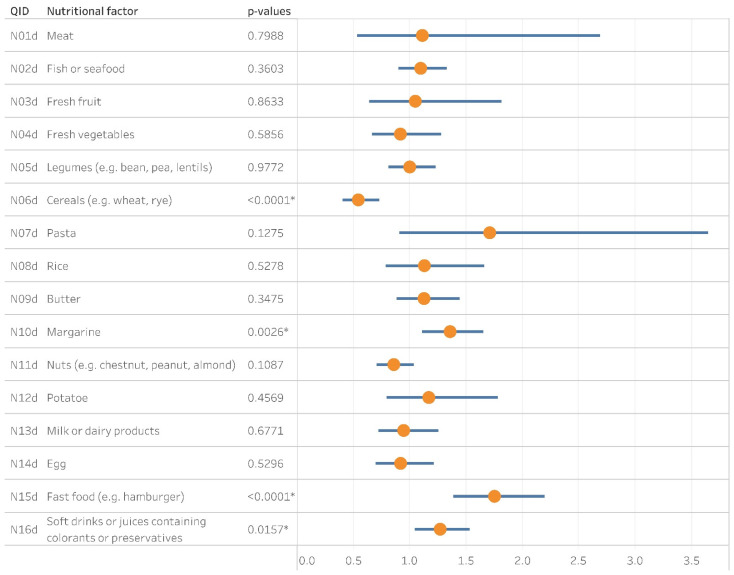
The Forest plot depicts the contribution of nutritional factors to the prevalence of cumulative asthma. QID: Question ID; Yellow dots and blue bars stand for the corresponding odds ratios and confidence intervals, respectively; *: *p* <0.05.

**Figure 5 ijerph-18-13403-f005:**
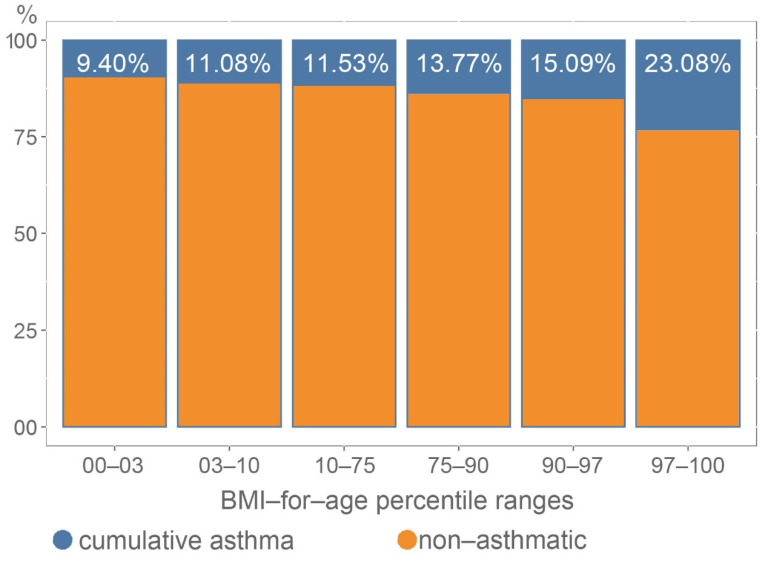
Association between cumulative asthma and BMI-for-age percentiles. (Chi-square = 16.26 df = 5 *p* = 0.0062).

**Table 1 ijerph-18-13403-t001:** The prevalence of asthma-related respiratory symptoms.

Has Your Child Ever Had Wheezing or Whistling in the Chest at Any Time in the Past?		
Yes	834	21.7%
No	3002	78.3%
**Has Your Child Had Wheezing or Whistling in the Chest in the Past 12 Months?**		
Yes	356	9.3%
No	3480	90.7%
**How Many Attacks of Wheezing Has Your Child Had in the Past 12 Months?**		
none	3598	93.8%
1–3	183	4.8%
4–12	42	1.1%
≥ 12	13	0.3%
**In the Past 12 Months, How Often, on Average, Has Your Child’s Sleep Been Disturbed Due to Wheezing?**		
Never woken with wheezing	3588	93.5%
Less than one night per week	162	4.2%
One or more nights per week	86	2.3%
**In the Past 12 Months, Has Wheezing Ever Been Severe Enough to Limit Your Child’s Speech to Only One or Two Words at a Time between Breaths?**		
Yes	41	1.1%
No	3795	98.9%
**Has Your Child Ever Had Asthma?**		
Yes	290	7.6%
No	3546	92.4%
**In the Past 12 Months, Has Your Child’s Chest Sounded Wheezy during or after Exercise? (Physical Education, Running, Walking on Stairs)**		
Yes	227	5.9%
No	3609	94.1%
**In the Past 12 Months, Has Your Child Had a Dry Cough at Night Apart from a Cough Associated with a Cold or Chest Infection?**		
Yes	422	11%
No	3414	89%
**Has Your Child Had Asthma Diagnosed by a Physician?**		
Yes	248	6.5%
No	3588	93.5%

**Table 2 ijerph-18-13403-t002:** The prevalence of current wheezing (CW), physician-diagnosed asthma (PDA) and cumulative asthma (CA) in the sample population.

Type of Asthma		Frequency	Percentage
Current wheezing (CW)	No	3480	90.7%
	Yes	356	9.3%
	Total	3836	100%
Physician-diagnosed asthma (PDA)	No	3588	93.5%
	Yes	248	6.5%
	Total	3836	100%
Cumulative asthma (CA)	No	3552	87.4%
	Yes	484	12.6%
	Total	3836	100%

**Table 3 ijerph-18-13403-t003:** The frequency of asthma symptoms indicates the severity in the CA group.

Severity Indicator		*n* in CA	% in CA
In the past 12 months, how often, on average, has your child’s sleep been disturbed due to wheezing?	Never woken with wheezing	292	60.33%
	Less than one night per week	119	24.59%
	One or more nights per week	73	15.01%
In the past 12 months, has wheezing ever been severe enough to limit your child’s speech to only one or two words at a time between breaths?	Yes	37	7.64%
	No	447	92.36%
In the past 12 months, has your child’s chest sounded wheezy during or after exercise? (physical education, running, walking on stairs)	Yes	149	30.79%
	No	335	69.21%

**Table 4 ijerph-18-13403-t004:** The association between CA asthma and other manifestations of atopy.

QID	Atopic Condition: Has Your Child Ever Had…?		*n* (%) in CA	*p*-Values	OR	CI
A1	any allergic diseases	Yes (*n* = 1206)	331 (27.45)	<0.0001	6.12	4.99–7.55
		No (*n* = 2630)	153 (5.82)			
A2	Eczema	Yes (*n* = 491)	86 (17.52)	0.0005	1.57	1.21–2.02
		No (*n* = 3345)	398 (11.90)			
A3	food allergy	Yes (*n* = 242)	61 (25.21)	<0.0001	2.53	1.85–3.42
		No (*n* = 3595)	423 (11.77)			
A4	allergic rhinitis	Yes (*n* = 357)	124 (34.73)	<0.0001	4.61	3.61–5.88
		No (*n* = 3479)	360 (10.35)			
A5	Has your child been diagnosed with allergic rhinitis by a physician?	Yes (*n* = 373)	149 (39.95)	<0.0001	6.21	4.90–7.86
		No (*n* = 3463)	335 (9.67)			

QID: question ID, OR: odds ratio, CI: confidence interval.

**Table 5 ijerph-18-13403-t005:** Statistical analysis of lifestyle factors; results revealed an inverse association between sports activity and cumulative asthma.

QID	Lifestyle Factors: Presence or Frequency of Behavioral Risk Factor during the Last 12 Months		*n* (%) in CA	*p*-Values	OR	CI
L1	Weekly frequency of sport activities?	≥3 times (*n* = 1879)	214 (11.39)	0.0098	0.69	0.52–0.92
		1–2 times (*n* = 1474)	194 (13.16)	0.1554	0.81	0.61–1.09
		Never or occasionally (*n* = 483)	76 (15.73)			
L2	Average time per week spent on watching TV and / or computer	≥3 h (*n* = 1948)	259 (13.30)	0.0965	1.39	0.96–2.08
		1–3 h (*n* = 1501)	185 (12.33)	0.2312	1.27	0.87–1.92
		<1 h (*n* = 322)	32 (9.94)			
		None responders (*n* = 65)	8 (12.31)			
L3	The child smokes occasionally or frequently	Yes (*n* = 4)	1 (25.00)	0.4687	2.31	0.11–18.10
		No (*n* = 3832)	483 (12.60)			

QID: question ID, OR: odds ratio, CI: confidence interval.

## Data Availability

Raw data are available from the corresponding author upon reasonable request. The datasets derived during this study are included in this published article (and its supplementary information files).

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
