# Peer review of "Prevalence of Asthma and Its Associating Environmental Factors among 6–12-Year-Old Schoolchildren in a Metropolitan Environment—A Cross-Sectional, Questionnaire-Based Study"

_ijerph, 2021, doi:10.3390/ijerph182413403_

Round 1

Reviewer 1 Report

In this study, the authors investigated the prevalence of current wheezing, physician-diagnosed asthma and the lifetime prevalence of asthma within a 6-12-year-old population in the urban environment of Budapest. They analyzed numerous environmental and lifestyle-related issues and their association with asthma. This information is valuable to be published.

The text requires several minor revisions:1. The abbreviation ISAAC in the abstract needs to be explained.2. The prevalence of current wheezing and physician-diagnosed asthma should be stated in the abstract3. In line 34. instead of “Air-pollution and a weedy area associated with greater risk” should stand “Air-pollution and a weedy area was associated with greater risk”4. Sentence in line 129: "In the case of categorical variables, odds ratios (OR) and 95% confidence intervals (95% CI) were calculated to establish how much more likely it was that someone who had the risk factor would develop asthma than someone who did not have it " should be written differently.

Author Response

Response to Reviewer #1 Comments

Dear Reviewer #1, we are thankful for your suggestions! We hope that the changes we have made will meet your expectations and improve the quality of the manuscript

  1. The abbreviation ISAAC in the abstract needs to be explained.

We replaced the abbreviation and used the whole term “International Study of Asthma and Allergies in Childhood”

  1. The prevalence of current wheezing and physician-diagnosed asthma should be stated in the abstract

We inserted our definition of cumulative asthma into the abstract. In addition, we indicated the prevalence of its compounds “Current wheezing and physician-diagnosed asthma showed a frequency of 9.5% and 6,3%, respectively.”

  1. In line 34. instead of “Air-pollution and a weedy area associated with greater risk” should stand “Air-pollution and a weedy area was associated with greater risk”

We rewrote the sentence and inserted the substantive verb.

  1. Sentence in line 129: "In the case of categorical variables, odds ratios (OR) and 95% confidence intervals (95% CI) were calculated to establish how much more likely it was that someone who had the risk factor would develop asthma than someone who did not have it " should be written differently.

We shortened the sentence to avoid any ambiguous expressions.

Reviewer 2 Report

IJERPH- 1497025

Comments and suggestions for authors

The paper by Dávid Molnár, Gabriella Gálffy et al. examines the prevalence of asthma and its associating environmental factors within a 6-12-year-old population of Hungarian youngsters. A cross-sectional, questionnaire-based study was conducted in primary schools located in the capital of Hungary.

The research topic is significant but it is not a novelty, however this type of research is characterized by the fact that it is hard to expect a large load of novelty from them.

As long as the assumptions if the International Study of Asthma and Allergies in Childhood (ISAAC) were referenced in the introduction, it begs to refer to at least a little bit more data and the effects obtained in the project. The authors are very sparing in words this means that, in principle, it is difficult to find the aim and the main assumptions of work, apart from the continuation of the research, which is assumed to be carried out once a decade. Nevertheless, the conduct of research itself cannot be a goal.

The results are neat, aesthetic and legible. Undoubtedly the strength of the publication is the size of tested population however, several aspects need to be corrected and explained in the publication. In the opinion of the reviewer, an excellent part of the discussion could be found in the introduction. This would explain the merits of dealing with the topic.

  • Line 49- The paragraph How to Use This Template should be deleted
  • Line 100- ‘We also included questions dealing with associating atopic conditions (A), including 100 food allergy, asthma and allergic rhinitis.’

The question is What was the structure of changed questionnaire? Whether it was somehow validated or someone else's previously described modifications to the form that authors used?

  • Line 102- What modifications to the base form were made to make it possible to monitor the nutrients’ influence? Was that 2003 created tool based on FFQ or some other tool? Nutrients monitoring is really demanding both from responders tool as well as for researchers. Appropriate developed and validated query is crucial.

  • Line 131- What do you mean by risk factors? From the abstract it is possible to predict but until now that even potential’ risk factors were not mentioned. Maybe the introduction is the place where the authors should signal what kind of risks were described until now in that term and which factors they will be discussing in the manuscript.

  • Line 126- Although the publication seems concise and clean, the authors are so sparing in words that the application of Individual regression models instead of multivariate is not clear in such a mass of variables.

  • Line 236- For children we rather talk about BMI z-scores standardized and calculated for age and gender.

  • The results of studies on the effects of pets are surprising. The pretty strong association of disease in dog owners and the not too strong but reduced rate among rodent owners is puzzling. Why rodents and not cats and dogs are less correlated with prevalence, although the hair and skin of these animals is less homologous to human hair. Our observations show that on the contrary, rodents, and especially feces and bedding, are more irritating. Cited cat paradox seems to be an intriguing phenomenon.

The main objections to the publication concern

  1. the lack of analysis of the influence of applied dietary supplements, which is one of the most important elements to consider when taking into account the nutritional factors.
  2. the lack of correlating the results with the family history of asthma.
  3. the presence of many typos mistakes and long sentence structures.

 After explaining the mentioned topics and text rewriting and correcting mistakes, it can be published.

Author Response

Response to Reviewer #2 Comments

Dear Reviewer #2, we really appreciate your valuable feedback regarding our manuscript. We tried our best to answer your questions and correct any errors to set our manuscript suitable for publication.

Unfortunately, only limited data is available from the study specific to our country's urban environment. Hungary participated with two centres in the phase three study of ISAAC in 2003, but none of them represented the capital. A survey (ISAAC ID 047002) covered schools of two cities from the southeastern lowland of Hungary, but even the more populated city of the two has a tenth of Budapest’s population and is surrounded by an agricultural area. There are limited data available from our country-specific results of the ISAAC studies. Most of them were published as parts of global reports, or minor results are available only in Hungarian (Zsigmond G et al., Gyermekgyógyászat, 2003 54:6). The meta-analysis of publicly available raw data of the ISAAC phases (Strachan D. et al. 2017, [data collection]. UK Data Service. SN: 8131,DOI: 10.5255/UKDA-SN-8131-1) was not our aim by this time.

According to our empiric observations and data requested from the National Health Insurance Fund of Hungary, asthma prevalence is currently increasing in such an urban environment. Inner cities are the areas where asthma is more frequent, this is also why we believe our cross-sectional study has a raison d’être. We applied the widely used ISAAC questionnaire, of which the last manual was published almost two decades ago. We finally made modifications to achieve our goals.

Our workgroup has recently published an article with the same methodology about the prevalence of allergic rhinitis (seen among references; Sultesz M et al, Allergy Asthma Clin Immunol. 2020;16(1):98) after almost a decade since our last epidemiological surveys (Sultész M et al., Allergol Immunopathol (Madr), 2017 45(5):487-495., Sultész M et al. Int J Pediatr Otorhinolaryngol, 2010 74(5):503-9.). This made us analyse both asthma and allergic rhinitis in the current project.

Line 49- The paragraph How to Use This Template should be deleted

Thank you for calling our attention to such a mistake! We deleted a remnant of the template.

Line 100- ‘We also included questions dealing with associating atopic conditions (A), including 100 food allergy, asthma and allergic rhinitis.’ The question is What was the structure of changed questionnaire? Whether it was somehow validated or someone else's previously described modifications to the form that authors used?

There is not a given, validated Hungarian questionnaire to approach the burden of atopic diseases. Thus, questions related to the environmental and nutritional factors and physical activity published in this study partially overlap with those found in the original ISAAC Phase two and three questionnaires.

We found other scientifically already discussed risk factors that had not been included in the ISAAC survey (for instance, the proximity of heavy traffic load and weedy areas). In our former publications (Sultész M et al., Allergol Immunopathol (Madr), 2017 45(5):487-495., Sultész M et al. Int J Pediatr Otorhinolaryngol, 2010 74(5):503-9.) we used a similar method, namely: the ISAAC core questions and the modified environmental and nutritional modules.

We also tried to get an overview of the general composition of the children’s meals. We used contemporary single word descriptions (e.g.: vegan, vegetarian etc.). We concluded that this chapter was not a sufficient tool because of the responders’ rate. Therefore, this modification was finally excluded from the study.

Questions assessing other possible associations with asthma, including the subjects’ family history, past medical history, medication and some socioeconomic factors, are not part of the current publication. These chapters are under evaluation and manuscript writing in a different context. As in several other studies, and in our subjects from a previous assessment, a family history of atopy correlated significantly with the prevalence of cumulative allergic rhinitis (Sultesz M et al, Allergy Asthma Clin Immunol. 2020;16(1):98)

Line 102- What modifications to the base form were made to make it possible to monitor the nutrients’ influence? Was that 2003 created tool based on FFQ or some other tool? Nutrients monitoring is really demanding both from responders tool as well as for researchers. Appropriate developed and validated query is crucial.

We agree that monitoring the influence of a given nutrient is a demanding question and its detailed analysis is beyond the scope of the current project. As discussed in the previous answer above, we tried our best to extend the ISAAC-based questionnaire, but some of the approaches failed, some delivered scientifically reasonable data. This cross-sectional questionnaire can serve descriptive association with asthma and nutrient consumption which covered the one-year period before the survey. This is one of the limitations of the publication that we updated in the corresponding section of the revised version.

Line 131- What do you mean by risk factors? From the abstract it is possible to predict but until now that even potential’ risk factors were not mentioned. Maybe the introduction is the place where the authors should signal what kind of risks were described until now in that term and which factors they will be discussing in the manuscript.

We briefly updated the last paragraph of the introduction with mentioning some examples.

Line 126- Although the publication seems concise and clean, the authors are so sparing in words that the application of Individual regression models instead of multivariate is not clear in such a mass of variables.

Our primary goal was to use descriptive statistics for the epidemiologic project. Because of the type of the survey (cross-sectional, questionnaire-based) and data collection we thought, that multivariable analyses wouldn’t be scientifically supported enough. We prefer multivariable analysis in randomised, controlled longitudinal clinical studies.

Line 236- For children we rather talk about BMI z-scores standardized and calculated for age and gender.

We used the guideline of the national authority of the Association of Hungarian Health Visitors as a methodological and an external data reference when calculated BMI-for-age. “…In clinical practice, BMI-for-age growth charts can be used to determine a child’s BMI-for-age percentile and to track relative weight status through childhood and adolescence. … In most research applications, either BMI z-scores or BMI-for-age percentiles can be used to determine cut points and classify weight status of children and adolescents”. (Must A & Anderson SE, Int J Obes (Lond). 2006 Apr;30(4):590-4. doi: 10.1038/sj.ijo.0803300.). The term “age-specific BMI” found in the methods section was corrected as „age and gender related body mass index (BMI) percentile (BMI-for-age)”.

The results of studies on the effects of pets are surprising. The pretty strong association of disease in dog owners and the not too strong but reduced rate among rodent owners is puzzling. Why rodents and not cats and dogs are less correlated with prevalence, although the hair and skin of these animals is less homologous to human hair. Our observations show that on the contrary, rodents, and especially feces and bedding, are more irritating. Cited cat paradox seems to be an intriguing phenomenon.

Thank you very much for sharing your personal experiences with us.

The main objections to the publication concern.

The lack of analysis of the influence of applied dietary supplements, which is one of the most important elements to consider when taking into account the nutritional factors

We hope that our answers above are welcome and acceptable. However, longitudinal surveys or twin studies in a bigger timeframe would be more appropriate tools to define etiological relationships between nutrients, dietary supplements, environmental factors, and asthma.

The lack of correlating the results with the family history of asthma.

As we mentioned above, the family history data is available, but it is part of a different project. Questions assessing family history, past medical history, medication and some socioeconomic factors are not part of the current publication. These results are under evaluation and manuscript writing in a different context.

The presence of many typos mistakes and long sentence structures.

We are really sorry if there are still typos in the manuscript despite all efforts and the external reviews. We have done another error check and asked a different native speaker (British English) for language editing. Errors have been corrected accordingly.

Round 2

Reviewer 2 Report

The manuscript can be Accepted in present form however some informations from the revieweres response would be welcomed in the introduction.